# Factors Affecting Return to Work of Patients with Chronic Coronary Syndrome: A Prospective Study

**DOI:** 10.3390/healthcare13182368

**Published:** 2025-09-20

**Authors:** Corina Oancea, Despina Mihaela Gherman, Rodica Simona Capraru, Sorina Maria Aurelian, Mirela Maria Nedelescu, Florina Georgeta Popescu

**Affiliations:** 1Faculty of Medicine, Carol Davila University of Medicine and Pharmacy, 020021 Bucharest, Romania; corina.oancea@umfcd.ro (C.O.); despina.gherman@umfcd.ro (D.M.G.); 2The National Institute for Medical Assessment and Work Capacity Rehabilitation, 050659 Bucharest, Romania; simona.capraru@cnpp.ro; 3Department of Occupational Health, Victor Babes University of Medicine and Pharmacy, 300041 Timisoara, Romania; popescu.florina@umft.ro

**Keywords:** chronic coronary syndrome, cardiac rehabilitation, predictor, return to work, employment

## Abstract

**Background/Objectives**: Return to work is an important goal of cardiac rehabilitation, yet individuals recovering from cardiovascular disease often face significant challenges in achieving it. As a result, a significant proportion of individuals with coronary artery disease experience work disability, negatively impacting both their economic well-being and quality of life while imposing a substantial financial burden on society. This less-studied issue is often treated as a secondary outcome in research, resulting in return to work findings that are frequently underreported. As such, this study aimed to identify the factors associated with adequate levels of functional capacity enabling the engagement in professional work and to develop a model for predicting the potential return to work of patients with coronary artery disease. **Methods:** During 2024, we enrolled 250 consecutive patients with chronic coronary syndrome less than 65 years of age who were referred to the National Institute for Medical Assessment and Work Capacity Rehabilitation (INEMRCM) for medical evaluation to establish eligibility-to-work disability benefits. Patients underwent a revascularization procedure either using PTCA or CABG, with a few having had no revascularization until the moment of assessment. Detailed demographic, socioeconomic, and clinical data were collected via interviews. Logistic regression was used to develop a multivariable model for predicting return to work. **Results:** Six months after discharge from the INEMRCM, around 20% of participants had returned to work. A better functional status was determinant for individuals’ re-employment (*p* = 0.026) along with an absence of cardiovascular comorbidities (*p* = 0.045) and holding a higher-skilled occupation (*p* = 0.037). The multivariate analysis identified professional specialization and the absence of cardiovascular comorbidities as the strongest predictors of return to work. **Conclusions:** Cardiac patients with coexisting cardiovascular conditions engaged in less-specialized types of work tend to experience poorer return to work outcomes. As such, individuals in this category should be carefully assessed and prioritized in the development of targeted rehabilitation programs.

## 1. Introduction

Ischemic heart disease (IHD) is one of the major causes of mortality, morbidity, and disability in the world; the World Health Organization found it to be responsible for 13% of total deaths globally [1]. Since 2000, a significant rise in mortality has been observed for this condition, with deaths increasing by 2.7 million to a total of 9.1 million in 2021 [2].

In 2023, data from the Romanian National Institute of Public Health indicated that coronary artery disease and hypertension along with diabetes were the three most chronic conditions, as they have consistently been in recent years [3].

IHD is the leading cause of mortality in Romania and has long been the main cause of work disability, only being surpassed in recent years by mental illness. This situation results in significant costs for individuals, with both repercussions for families and direct and indirect social costs [4].

Cardiovascular disease (CVD) leads to significant productivity losses due to premature death, a diminished work capacity, absenteeism, and early retirement among affected individuals. The economic impact of CVD in Europe is estimated at EUR 282 billion per year, with a substantial share attributed to these productivity losses [5].

Even under these circumstances, the introduction of new treatment regimens for acute management and primary and secondary prevention has improved the prognosis for acute coronary events. This has led to an increasing number of survivors returning to work after treatment.

Cardiac rehabilitation (CR) aims to stabilize or slow the progression of cardiovascular disease while supporting patients in resuming an active lifestyle. Its core components are well-established and include physical activity recommendations, supervised exercise training, dietary counseling, weight loss, increased adherence to specific medication, smoking cessation, diabetes control, psychological support, and behavioral therapy [6,7].

CR is a first-class recommendation in all current clinical guidelines for patients with stable angina or myocardial infarction or for those who have undergone coronary revascularization (either percutaneous or surgical). These recommendations are strongly supported by evidence from large-scale studies, randomized controlled trials, and meta-analyses involving patients with cardiovascular conditions [8].

There is growing evidence on cardiac rehabilitation in Romania. A survey from 2024 found a 69% increase in Romanian cardiac rehabilitation centers since 2017 but noted a geographic imbalance, mostly urban/private, while telerehabilitation is still rare [9]. More research is needed to better understand the impact of chronic coronary syndromes on individuals’ ability to continue working.

Our aim was to identify potential factors that influence return to work among patients with chronic coronary syndrome, as understanding the potential risks may be of crucial importance to design personalized rehabilitation programs for individuals at increased risk of future work disability after acute myocardial infarction.

To address this, our research was structured into two phases. The first phase focused on identifying factors associated with the degree of functional impairment that qualifies patients for social insurance benefits, as defined by the current eligibility criteria in Romania [10]. The second phase assessed the relationship between these factors and employment status by evaluating the incidence of return to work at the 6-month follow-up.

## 2. Materials and Methods

### 2.1. Study Design and Population

Between 1 January 2024 and 31 December 2024, a prospective cohort study was conducted involving 250 consecutive patients. The inclusion criteria for study participants were age under 65 years, application for work disability benefits, and assessment for eligibility at INEMRCM. Each patient had been diagnosed with chronic coronary syndrome by a cardiologist, following the European Cardiac Society’s criteria [11]. Patients aged either under 18 years or over 65 years or those with outdated contact information at the 6-month follow-up were excluded.

The analysis of cases was conducted at distance after the acute event, either involving individuals who were applying for a disability pension after six months to one year of sick leave or those who had already been receiving a work disability pension for at least one year.

None had participated in cardiac rehabilitation following an acute coronary syndrome (ACS) prior to their admission to INEMRCM. The most commonly reported reasons for non-participation included low compliance, a lack of understanding of the importance of cardiac rehabilitation, the distance and inaccessibility of rehabilitation centers, or being part of disadvantaged groups without public insurance, which made them ineligible for the program.

According to established treatment, patients benefit from a revascularization procedure either using PTCA or CABG, with a few having had no revascularization until the moment of assessment. Detailed demographic, socioeconomic, and clinical data were collected via interview.

### 2.2. Data Collection

Demographic data included age, sex, residence, education level, and occupational group. Occupation was classified according to the Romanian version of the International Standard for the Classification of Occupations—ISCO/Romanian Classification of Occupations (COR)—into one of nine classes: managers; professionals; technicians and associate professionals; clerical support workers; service and sales workers; skilled agricultural, forestry, and fishery workers; craft and related trades workers; plant and machine operators; and assemblers and elementary occupations. Larger professional categories were distinguished as manual/semi-professional (specialized) and professional labor. Based on the typical educational requirements for different occupations, three main categories of education were analyzed: low, medium, and high.

Socioeconomic data were registered as follows: professional status before assessment (with the following categories: unemployed/prolonged sick leave or benefit from one of the degrees of work disability), number of dependents (children under age 18 in family care), length of service duration, and professional route recording the number of previous jobs.

Work disability was classified according to Law 360/2023 on the public pension system based on the extent of functional loss as follows: 1st degree—characterized by very severe functional impairment and a complete loss of self-care abilities, requiring daily assistance for basic activities; 2nd degree—defined by severe functional impairment with partially preserved ability to take care of oneself; and 3rd degree—involving moderate functional impairment with preserved self-care [12].

Third-degree work incapacity, characterized by moderate impairment, often allows normal professional activity, typically on a part-time basis. In contrast, individuals in the first- and second-degree categories with very severe or severe impairments can only work under specific conditions, such as reduced daily hours, flexible hours, or work from home.

Clinical data were noted as the severity of angina according to the Canadian Cardiovascular Society (CCS) Angina Grading Scale [13] and resting ECG changes and ejection fraction as indicators of cardiac function at rest. Ejection fraction was analyzed according to three major categories as per the ESC guidelines 2023: reduced LV ejection fraction less than 40%, mildly reduced between 41% and 49%, and normal above 50% [14]. The following additional clinical data were also included: exercise capacity expressed in terms of metabolic equivalents (METs), number of comorbidities, cardiovascular risk factors (hypertension, obesity, dyslipidemia, type 2 diabetes), associated CVD (mainly carotid artery disease or peripheral artery disease), mental disorder certified by a psychiatrist, severity of the disease according to the number of involved major coronary arteries from coronary angiography performed prior to admission to INEMRCM, type of treatment (PTCA, CABG, or medical), and success of treatment (partial or total revascularization).

### 2.3. Exercise Testing

Exercise testing is the most utilized method for evaluating functional capacity, assessing coronary risk, and determining prognosis. An individualized ramp cycle ergometer protocol was used to assess exercise capacity. Ramp protocols provide a more gradual and patient-friendly alternative to treadmill testing, minimizing the abrupt and rapid changes in workload characteristic of commonly used step protocols such as the Bruce protocol [15].

Estimated METs were expressed as a percentage of the predicted values based on age, sex, and height. Exercise capacity was assessed according to the Goldman capacity scale and expressed in terms of metabolic equivalents (METs) in 4 categories: 1st class—patient can perform activities requiring ≤2 METs to completion; 2nd class—patient can perform activities requiring 3–4 METs to completion; 3rd class—patient can perform activities requiring 5–6 METs to completion; and 4th class—patient can perform activities requiring ≥7 METs to completion [16].

### 2.4. Follow-Up

Six months after hospitalization, a social worker contacted the patients by telephone who were assessed as having adequate work capacity to safely participate in employment. Patients were asked if they had returned to work and, if applicable, how many hours per day they were working. Return to work (RTW) was defined as the resumption of paid employment either on a part-time or full-time basis (defined as working either 4 or 8 h per day).

Two outcomes were considered in this study:-‘Estimated work capacity’, assessed at the institute through functional tests as an indicator of overall health status and used to determine the degree of work disability. This outcome corresponds to the degree of determined functional impairment based on current medical criteria for granting 1st, 2nd, or 3rd degree of work disability pensions. Functional status was also analyzed as a dichotomous dependent variable, with ‘no reduction’ as the reference group, to assess the independent impact of various variables that significantly influenced estimated work ability in the descriptive statistics.-‘Employment/return to work’, representing the self-reported situation of actual paid employment.

Statistical analysis: After verifying the normality of the data distribution, statistical analysis was conducted using PSPP.4 software (https://www.gnu.org/software/pspp/get.html, 7 January 2025). A *p*-value of less than 0.05 was considered statistically significant, with results reported at a 95% confidence interval. All demographic, socioeconomic, and medical parameters were compared between groups using a chi-square test (for nominal and categorical variables) or an independent-samples *t*-test (for numeric variables). Work status (disabled vs. active) was utilized as the dependent variable to evaluate each factor as a potential predictor of work disability. Logistic regression was used to construct a multivariable model for predicting return to work and to calculate odds ratios (ORs), quantifying the strength and nature of the associations between different variables and work capacity.

## 3. Results

### 3.1. Primary Evaluation

The baseline characteristics of patients with chronic coronary syndrome are presented in Table 1.

Comparing the patients’ status at admission to the INEMRCM, there were significant differences between groups, analyzed as a dichotomized variable (work-disabled vs. unemployed/prolonged sick leave), and between all categories (no or 1st, 2nd, or 3rd degree of work disability).

Patients not yet classified for work disability were significantly younger: 53.28 ± 5.43 vs. 54.56 ± 4.63; *p* = 0.049. The difference also remained significant after the evaluation: patients considered not disabled were significantly younger than those granted a degree of work disability—53.06 ± 5.07 vs. 55.10 ± 4.68; *p* = 0.001.

The difference also remained significant when calculated for all categories; *p* = 0.002. As people age, the risks associated with cardiovascular disease increase, including its functional impact [17].

There were significant differences regarding associated risk factors: patients with hypertension were older (54.70 ± 4.42 vs. 52.46 ± 5.28; *p* = 0.05), while those with obesity were younger (53.39 ± 4.67 vs. 55.14 ± 4.44; *p* = 0.020). There were no significant differences between groups for the other cardiovascular risk factors, such as dyslipidemia or diabetes, or other associated cardiovascular diseases.

Older patients had either a reduced or preserved ejection fraction, while most younger patients had a mildly reduced EF (*p* = 0.015) and poorer functional capacity expressed in METs during the effort testing (*p* = 0.052). The left ventricular ejection fraction remains the major parameter for diagnosis, prognosis, and treatment decisions in heart failure.

Patients with a preserved work capacity displayed the following characteristics: mainly men (*p* = 0.029), more educated (close to statistical significance; *p* = 0.060), less severe angina (*p* = 0.001), without ECG changes (*p* = 0.001), more with full revascularization (*p* < 0.001), less cardiovascular-associated pathologies (*p* = 0.003), better functional status/more METs (*p* < 0.001), and a more preserved ejection fraction (*p* < 0.001) (Table 2).

Using logistic regression, we investigated the influence of each factor on the estimated work capacity. We initially introduced all variables that were significantly correlated with work capacity—specifically the gender, level of education, angina, ECG, treatment success, cardiovascular comorbidities, exercise capacity, and ejection fraction. Through the backward selection process, we eliminated the variables that lost statistical significance in the multivariate model. The R-squared value indicated that the model was reasonably effective in predicting the factors associated with a better work capacity (Table 3). The multivariate analysis suggests that the male sex, absence of ECG changes, and good exercise level (in METs) may serve as predictors of a high likelihood of employment in cases of coronary artery disease.

### 3.2. Results of the Follow-Up

At the 6-month follow-up phone call, the non-response rate was high at 50.70% (109 persons). Thirty-five individuals were excluded from the follow-up analysis due to missing or outdated contact information. The status of the 106 individuals analyzed was as follows: 29 (27.36%) had chosen not to work, 22 (20.75%) were employed—3 working 4 h per day and 19 working 8 h per day, 42 (39.62%) were solely dependent on disability pension incomes, 12 (11.32%) had applied for early or old-age retirement, and 1 participant (0.94%) was deceased due to severe complications from peripheral artery disease in the lower limbs.

There were no significant differences between respondents and non-respondents regarding their age, sex, education level, occupation type, disease severity, comorbidities, treatment type, or therapeutic success. The assessment of the work capacity upon discharge from the INEMRCM also showed no significant difference between the two groups, with a preserved work capacity reported in 50.46% of non-respondents and 50.35% of respondents. This means that, in theory, their health condition would allow them to safely engage in professional activities in line with current cardiology guidelines and national standards for social security benefit eligibility.

However, non-respondents were significantly more likely to reside in rural areas (64 individuals, 58.72%; *p* = 0.041), none experienced angina at low exercise thresholds compared to 10 respondents who did (7.09%; *p* = 0.028), and a greater proportion demonstrated a higher exercise capacity (≥7 METs: 36.76% vs. 31.58%; *p* = 0.037).

The following variables, shown in Table 4, emerged as significant for resuming work: cardiovascular comorbidities (*p* = 0.045), effort capacity (*p* = 0.026), professional category (*p* = 0.037), and age (approaching statistical significance; *p* = 0.055).

The backward logistic regression analysis confirms that holding a more specialized occupation and the absence of other cardiovascular pathologies increase the likelihood of employment in case of chronic coronary syndrome (Table 5).

## 4. Discussion

Our study aimed to assess the factors influencing the work capacity in 250 patients diagnosed with coronary artery disease (CAD). The strongest predictors of a better work capacity were sex, the absence of ECG abnormalities, and a high level of physical activity. Additionally, individuals who returned to work were more likely to hold higher-skilled occupations and have no additional cardiovascular comorbidities.

### 4.1. Sex, Coronary Artery Disease, and Work

Males are more likely to return to work, which may be partly explained by their historically longer and more continuous working experience due to societal roles and expectations. Several social, cultural, and economic factors may influence the length of an individual’s working life. In many societies, traditional gender roles have assigned men the primary role of breadwinners, often resulting in men having longer and continuous work histories [18].

Earlier studies have demonstrated that return to work rates following myocardial infarction or interventional procedures are considerably lower in women compared to men [19,20]. However, data on women’s return to work after cardiovascular events remain limited. More recent studies continue to report lower recovery rates among women [21,22]. Cardiac rehabilitation outcomes also indicate lower participation, reduced adherence, and significantly higher dropout rates in women; although, those who complete the rehabilitation program achieve functional improvements comparable to or exceeding those observed in men.

### 4.2. Age, Exercise Capacity, Coronary Artery Disease, and Work

Older adults tend to have decreased levels of physical activity, given that aging is characterized by a progressive and cumulative generalized impairment of physiological function [23]. The precise impact of aging on ventricular function has remained controversial. Several studies involving different methods concluded that aging does not have any impact on LV function, whereas others found a slightly decreased LV ejection fraction (LV-EF) in healthy subjects [24].

Regular physical activity is widely recognized as a key factor in improving hemodynamic parameters, as well as the management and prevention of cardiovascular disease (CVD) and arterial aging [25]. In clinical assessments, the exercise capacity expressed in metabolic equivalents (METs) provides an objective measure of functional reserve.

An exercise capacity exceeding seven metabolic equivalents (METs) is associated with the ability to safely perform certain physically demanding professions. Research has shown that an exercise capacity of seven METs or higher is linked to lower rates of cardiovascular events and mortality. This indicates a level of physical fitness sufficient for moderately intense activities, but job-specific requirements and individual factors should also be considered [26]. In our study, a higher exercise capacity was significantly associated with return to work.

These findings strengthen the importance of incorporating exercise testing into rehabilitation programs, both for clinical risk stratification and for guiding vocational counseling [27].

### 4.3. Resting ECG, Coronary Artery Disease, and Work

A resting electrocardiogram (ECG) is commonly performed for cardiovascular disease (CVD)-screening purposes/early prevention of CVD events in working populations [27]. In our study, the absence of ECG abnormalities was a significant predictor of preserved work capacity, highlighting its value not only as a diagnostic marker but also as a functional indicator relevant to occupational outcomes. These findings support the use of ECG in the comprehensive assessment of patients with chronic coronary syndrome, where it may help identify individuals with a higher likelihood of successful reintegration into employment.

### 4.4. Associated Cardiovascular Diseases and Work

Atherosclerosis, the underlying process of coronary artery disease, frequently leads to the development of additional cardiovascular conditions such as peripheral artery disease or carotid artery disease. These comorbidities share common inflammatory mechanisms and contribute to cumulative vascular damage, which in turn worsens functional capacity [28]. In our study, the absence of such comorbidities emerged as one of the strongest predictors of return to work, underscoring the impact of the systemic atherosclerotic burden on occupational outcomes. Patients with a coexisting vascular disease are more likely to experience impaired exercise tolerance, greater disability, and reduced chances of reintegration into the workforce.

In a long-term prospective study, Zierfuss and colleagues found that patients with coexisting cardiovascular conditions—such as peripheral arterial disease (PAD) and extracranial carotid artery disease (ECAD)—faced significantly elevated rates of fatal cardiovascular events. They emphasized that this group constitutes a particularly vulnerable population and advocated for more thorough evaluations, intensified monitoring, and enhanced risk reduction strategies to address both all-cause and cardiovascular mortality [29].

Carlsson et al., in a retrospective study involving over 15,000 individuals from the Swedish national registry, demonstrated that peripheral artery disease, along with stroke and myocardial infarction, was linked to a higher risk of early retirement, cardiovascular events, and mortality [30].

These findings highlight the importance of the early identification and management of cardiovascular comorbidities in order to improve both the health prognosis and work-related rehabilitation.

### 4.5. Type of Occupation and Work

The literature provides extensive data on the impact of socio-professional factors on return to work (RTW), including in the context of cardiovascular disease. Cancelliere conducted a synthesis of systematic reviews to identify common prognostic factors for return to work across various health and injury conditions. Among the numerous factors evaluated, only a few were consistently associated with return to work across conditions, including a higher education and socioeconomic status, a lower severity of injury or illness, return to work coordination, and multidisciplinary interventions involving the workplace and relevant stakeholders [31].

A meta-analysis of prospective studies conducted by S.H.Y. Kai et al. demonstrated that a lower level of education significantly reduced the RTW prevalence, while higher rates were observed among white-collar workers and individuals with less physical job demands. A similar finding—showing a higher prevalence of return to work among white-collar workers compared to blue-collar workers—was also reported by Sadeghi et al. in a separate systematic review and meta-analysis, with several other studies supporting this trend [32,33]. Occupational physical constraints appear to negatively affect RTW. Gaining a better understanding of real-world working conditions that influence RTW would be valuable for supporting the continued employment of coronary patients [34].

### 4.6. Employment in Coronary Artery Disease

This prospective study also examined the possibilities of workforce reintegration among its cardiac patients. It was found that approximately one-quarter of the participants returned to work at least one year after experiencing an acute myocardial infarction. The strongest predictors for returning to work appeared to be the type of occupation and the absence of additional cardiovascular comorbidities. Between these two factors, the occupation had the highest odds ratio, indicating a strong association with the resumption of employment.

Other studies have indicated that return to work is influenced more by socio-occupational factors than clinical conditions [35]. A study conducted by Wang and colleagues found that individuals with lower levels of education are at a higher risk of permanent work disability following an acute myocardial infarction [36]. Furthermore, Danchin and Goepfert suggested that psychosocial factors may have a greater influence on return to work (RTW) than exercise training alone, based on their evaluation of the effect of exercise training and cardiac rehabilitation in men under 60 years with acute myocardial infarction. They found no significant difference in RTW between patients undergoing comprehensive rehabilitation and those receiving standard care, and after three years, employment rates were even lower in the rehabilitation group [37].

From a clinical perspective, patients with coronary artery disease frequently present other cardiovascular conditions, such as cerebrovascular disease or peripheral artery disease, as they share atherosclerosis as a common underlying cause [38].

In this study, the absence of additional cardiovascular comorbidities was associated with a lower risk of unemployment following acute myocardial infarction. This finding suggests that coexisting cardiovascular conditions may worsen the prognosis of coronary artery disease with respect to work disability. It is important to note that patients experiencing complications of atherosclerosis, such as stroke or advanced peripheral artery disease, are at high risk of functional impairment [36]. Nevertheless, the prognosis can be improved through secondary preventive measures, medical treatment, lifestyle changes, and the control of modifiable CV risk factors.

Unlike most reports in the literature, the proportion of patients who resumed employment in our study was comparatively low. Two meta-analyses estimated an overall return to work prevalence of approximately 80% after a coronary event, with rates ranging from 41.2% to 100% [34,39]. A plausible explanation for our lower figure is the longer interval between the coronary event and the follow-up assessment, as previous studies have shown that the probability of workforce reintegration declines with time.

Positive outcomes are clear at three months in patients who receive cardiac rehabilitation; however, long-term adherence to secondary prevention guidelines tends to decline, leading to a gradual loss of functional capacity. This is reinforced by a 2023 study conducted by researchers from the ‘Victor Babeș’ University in Timișoara who monitored 480 patients post-coronary syndrome over a 12-month period and found that participation in the cardiac rehabilitation program was low, with only approximately 27% completing the entire program [40].

Furthermore, the probability of returning to work after being granted a long-term disability benefit is below 2% annually on average, meaning that for the vast majority of beneficiaries, work disability effectively marks the end of their working life [41].

Additionally, these patients did not undergo cardiovascular rehabilitation, which likely reduced their chances of early reintegration. A study by Lamberti et al. demonstrated that non-participation in cardiac rehabilitation was a significant contributor to poorer occupational outcomes. The authors emphasized the crucial role of cardiac rehabilitation and occupational counseling in facilitating recovery and successful workplace reintegration [42]. In this context, patient education and enhanced empowerment—supported by emerging telerehabilitation and remote recovery approaches—may contribute to sustained benefits.

There is considerable overlap between the factors involved in both the estimated and real-life rehabilitation of work capacity. Among the key medical determinants, the residual functional capacity—effectively evaluated through stress testing—and the presence of coexisting cardiovascular conditions stand out as the most critical. In terms of non-medical influences, as numerous studies have highlighted, the nature of the work involved is significant.

These findings emphasize the need for a more nuanced approach to assessing work capacity in patients with chronic coronary syndrome. It is essential to recognize the compounding negative impact of additional cardiovascular diseases on functional status and urgently integrate vocational rehabilitation into the process.

Strength of the study: We carried out a prospective study and simultaneously examined the impact of several types of predictors of work resumption in patients with chronic coronary syndrome: namely demographic, socioeconomic, and clinical predictors. Most studies have examined return to work within a short period following the cardiac event. Fewer studies have assessed the work status over a longer duration after acute myocardial infarction; therefore, our research provides valuable additional insights in this area.

Study limitations: The sample size was small and recruited from a single hospital; thus, results cannot be considered as representative for all patients with chronic coronary syndrome and should be interpreted with caution. The nature of the hospital introduces a selection bias, as this study reflects a group of patients primarily from disadvantaged areas, with lower educational levels and a greater reliance on social support. It can be assumed that individuals with higher education levels and more favorable socioeconomic backgrounds may have more positive work experiences and a higher likelihood of returning to work.

Our study group was also heterogeneous, with considerable variability in the time elapsed since participants experienced an acute coronary event. While this introduces a potential confounding factor that complicates the identification of the true effects of the variables studied, it also enhances the generalizability of the findings to a broader population.

The inability to contact a substantial proportion of patients reduced the statistical power of this study and limited the precision with which both the return to work rate and its predictors could be assessed. This also introduced the risk of a non-response bias. Notably, non-respondents were more often rural residents, which may reflect either reluctance or barriers to communication access. In addition, their more favorable clinical profile (absence of angina at low exertion and better exercise capacity) suggests that they may have had a better functional status and work capacity. If so, the true rate of return to work may have been underestimated in our results. While this remains speculative, future studies should address missing data more systematically, for example, through sensitivity analyses.

Further directions for research: Larger studies on a homogenous group of subjects would better describe the relationship between different factors and return to work. Collecting more detailed information about patients’ circumstances and return to work outcomes would provide a broader understanding of the situation. Additionally, comparing these findings with patients who have undergone cardiac rehabilitation would offer a clearer picture of the benefits of such interventions and support the development of more effective programs.

## 5. Conclusions

In this prospective cohort of patients with chronic coronary syndrome, only one-fifth returned to work within six months, despite nearly half demonstrating preserved functional capacity. A better exercise tolerance, the absence of cardiovascular-associated pathologies, and engagement in specialized or professional occupations were the main determinants of successful reintegration. The multivariate analysis confirmed occupational specialization and the absence of comorbid cardiovascular diseases as the strongest independent predictors. These findings underscore that returning to work after coronary events without adequate support is highly challenging. To improve outcomes, a multidisciplinary approach is necessary, along with the development of effective mechanisms to facilitate employment—such as supportive legislation, clear methodologies, and dedicated rehabilitation structures.

## Figures and Tables

**Table 1 healthcare-13-02368-t001:** Main demographic, socioeconomic, and clinical characteristics of participants.

Characteristics	Values ± SD/Number (%)
Mean age (years)	54.08 ± 4.98
Gender (male)	198 (79.20%)
Residence (urban)	120 (48%)
Mean work experience (years)	20.23 ± 9.98
Level of education	
-low	100 (40.49%)
-medium	140 (56.68%)
-high	7 (2.83%)
Occupation	
-manual labor	172 (75.77%)
-specialized labor	48 (21.15%)
-professional labor	7 (3.08%)
Career change during working life (yes)	93 (37.80%)
Angina	
-no angina	20 (8.03%)
-high effort	23 (9.24%)
-mid effort	196 (78.71%)
-low effort	10 (4.02%)
Resting ECG changes (yes)	187 (75.10%)
Ejection fraction of left ventricle	
-reduced ≤ 40%	31 (12.45%)
-mildly reduced 41–49%	27 (10.84%)
-preserved ≥ 50%	191 (76.71%)
Exercise capacity (METs)	
1st class: activities ≤ 2 METs	11 (6.75%)
2nd class: activities 3–4 METs	21 (12.88%)
3rd class: activities 5–6 METs	76 (46.63%)
4th class: activities ≥ 7 METs	55 (33.74%)
Comorbidities	
-absent	6 (2.40%)
-1 or 2	124 (49.60%)
-3 or 4	100 (40%)
-5 or more	20 (8%)
Hypertension (yes)	122 (82.43%)
Obesity (yes)	72 (48.32%)
Dyslipidemia (yes)	84 (56.38%)
Type 2 diabetes (yes)	46 (31.08%)
Mental disorders (yes)	11 (7.38%)
Severity of disease	
-no coronary artery involved	8 (5.67%)
-one coronary artery	60 (42.55%)
-two coronary arteries	46 (32.62%)
-≥three coronary arteries	27 (19.15%)
Type of treatment	
-conservative	22 (15.60%)
-PTCA	109 (77.30%)
-CABG	10 (7.09%)
Treatment success—coronary angiography	
-not performed	11 (4.40%)
-no lesions/solved lesions	160 (64%)
-no solved lesions	79 (31.60%)

**Table 2 healthcare-13-02368-t002:** Factors influencing estimated work capacity.

Characteristics	Fit for Work	Significance (*p*)
Gender (male/female)—Yes (%)	54.04%/36.54%	0.029
Level of education	Yes/No (%)	0.060
-low	33.60%/47.54%	
-medium	62.40%/50.82%	
-high	4.00%/1.64%	
Angina	Yes/No (%)	0.001
-no angina	11.11%/4.88%	
-high effort	15.08%/3.25%	
-mid effort	72.22%/85.37%	
-low effort	1.59%/6.50%	
Resting ECG changes—Yes/No (%)	43.85%/69.35%	0.001
Ejection fraction of left ventricle	Yes/No (%)	<0.001
-reduced ≤ 40%	0.00%/25.20%	
-mildly reduced 41–49%	4.76%/17.07%	
-preserved ≥ 50%	95.24%/57.72%	
Exercise capacity (METs)	Yes/No (%)	<0.001
1st class: activities ≤ 2 METs	0.83%/23.26%	
2nd class: activities 3–4 METs	7.50%/27.91%	
3rd class: activities 5–6 METs	48.33%/41.86%	
4th class: activities ≥ 7 METs	43.33%/6.98%	
Cardiovascular-associated pathologies	31.03%/63.33%	0.003
Yes/No (%)		
Treatment success—coronary angiography	76.98%/20.63%	<0.001
-no lesions or solved lesions vs. no solved lesions—Yes (%)		

**Table 3 healthcare-13-02368-t003:** Logistic regression analysis with predictive factors for work capacity (R-squared = 0.43).

Variable	B Coefficient	Standard Error	Significance (*p*)	OR (95%CI)
Constant	1.62	1.20	=0.176	
Sex	1.00	0.50	=0.046	2.72 (1.02–7.28)
Abnormal resting ECG	1.43	0.55	=0.009	4.19 (1.43–12.29)
Exercise capacity	−1.71	0.33	<0.001	0.18 (0.09–0.34)

**Table 4 healthcare-13-02368-t004:** Factors associated with workforce reintegration.

Characteristics	Fit for Work	Significance (*p*)
Type of occupation	Yes/No (%)	0.037
-professional work	0.00%/3.66%	
-specialized jobs	40.91%/18.32%	
-manual work	59.09%/78.01%	
Age group	Yes/No (%)	0.055
1. 41–50 years	40.91%/21.24%	
2. 51–60 years	59.09%/68.39%	
3. >61 years	0.00%/10.36%	
Exercise capacity (METs)	Yes/No (%)	0.026
1st class: activities ≤ 2 METs	5.56%/0.00%	
2nd class: activities 3–4 METs	0.00%/14.53%	
3rd class: activities 5–6 METs	55.56%/50.43%	
4th class: activities ≥ 7 METs	38.89%/35.04%	
Cardiovascular-associated pathologies	20.00%/39.55%	0.045
Yes/No (%)		

**Table 5 healthcare-13-02368-t005:** Logistic regression analysis with predictive factors for re-employment (R-squared = 0.11).

Variable	B Coefficient	Standard Error	Significance (*p*)	OR (95%CI)
Constant	0.19	1.27	=0.881	
Occupation	1.03	0.49	=0.036	2.80 (1.07–7.34)
C-v comorbidities	1.33	0.60	=0.026	0.26 (0.08–0.85)

## Data Availability

The data presented in this study are available on reasonable request from the corresponding author.

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
