# Peer review of "Factors Affecting Return to Work of Patients with Chronic Coronary Syndrome: A Prospective Study"

_healthcare, 2025, doi:10.3390/healthcare13182368_

Round 1
Reviewer 1 Report
Comments and Suggestions for Authors
Congratulations for your work. The manuscript is well-written, although the absence of subtitles in some parts decreases comprehension. The structure should be presented more clearly and some parts should be shortened as they are too extended and provide inappropriate information. Some suggestions for improvement:
- Introduction: Some information is irrelevant (for example the sentence for COVID-19 pandemic used probably for contrast, which is not described properly, making this sentence unnecessary). Some parts like the work ability categorization is unnecessary. This should be presented in the methods part if this categorization is necessary for the description of functional impairment in your results. Generally, introduction should be shortened.
- Methods: Except for outcomes and statistical analysis, the methods are preseneted in a long manuscript without proper structure. Please separate paragraphs and subtitles like study design, study population, inclusion/exclusion criteria, description of the variables, primary and secondary outcomes, subgroup analysis. By separating these paragraphs you will offer a better structure to the manuscript and you will find some parts, which probably are not necessary for your results.
- Results: This part is the most interesting and is well-presented. It looks like another authors has written this part and other authors the rest of the manuscript.
- Discussion: It is way too long and be shortened. Some parts are well-written, where you comment your findings and compare them to existing literature. This is the aim of the discussion in an original article. However, some parts present irrelevant information to your findings and describe issues without even a citation (two paragraphs in 4.1 section for example, and other parts too), while these parts do not describe your findings. These parts should be omitted. In some parts you present studies like a well narrative review, while you do not present any comparison to your findings. Probably you should present these findings in another manuscript whose aim would be to present a review of the literature. Please try to keep only the studies which present relevant results to your findings and make the comparison if there any. In this case, you will avoid all inappropriate information and discussion will the a proper extent avoiding also some repetitions expessed in different words.
Author Response
Dear member of the reviewer board,
Thank you for reviewing our article submitted for publication. We appreciate your insightful comments and have addressed them by making several revisions, as outlined below.
Sincerely,
Dr. Corina Oancea.
- Introduction: Some information is irrelevant (for example the sentence for COVID-19 pandemic used probably for contrast, which is not described properly, making this sentence unnecessary). Some parts like the work ability categorization is unnecessary. This should be presented in the methods part if this categorization is necessary for the description of functional impairment in your results. Generally, introduction should be shortened.
A: We deleted the sentence about Covid-19 and other parts from Introduction and moved the categories of work ability in the Methods part.
- Methods: Except for outcomes and statistical analysis, the methods are presented in a long manuscript without proper structure. Please separate paragraphs and subtitles like study design, study population, inclusion/exclusion criteria, description of the variables, primary and secondary outcomes, subgroup analysis. By separating these paragraphs you will offer a better structure to the manuscript and you will find some parts, which probably are not necessary for your results.
A: We introduced subtitles: study design and population with inclusion / exclusion criteria, data collection, exercise testing and follow-up.
- Results: This part is the most interesting and is well-presented. It looks like another authors has written this part and other authors the rest of the manuscript.
- Discussion: It is way too long and be shortened. Some parts are well-written, where you comment your findings and compare them to existing literature. This is the aim of the discussion in an original article. However, some parts present irrelevant information to your findings and describe issues without even a citation (two paragraphs in 4.1 section for example, and other parts too), while these parts do not describe your findings. These parts should be omitted. In some parts you present studies like a well narrative review, while you do not present any comparison to your findings. Probably you should present these findings in another manuscript whose aim would be to present a review of the literature. Please try to keep only the studies which present relevant results to your findings and make the comparison if there any. In this case, you will avoid all inappropriate information and discussion will the a proper extent avoiding also some repetitions expessed in different words.
A: We shortened the discussion part.
Reviewer 2 Report
Comments and Suggestions for Authors
-
Revise to address the high non-response rate with sensitivity analyses or additional discussion of potential bias.
-
Enhance the Discussion by tightening sections less directly related to core findings (e.g., wearable devices).
-
Copyedit to correct typographical errors and improve English usage throughout.
Author Response
Dear member of the reviewer board,
Thank you for reviewing our article submitted for publication. We appreciate your insightful comments and have addressed them by making several revisions, as outlined below.
Sincerely,
Dr. Corina Oancea.
- Revise to address the high non-response rate with sensitivity analyses or additional discussion of potential bias.
A: We rewrote and strengthen the limitation section:
The inability to contact a substantial proportion of patients reduced the statistical power of the study and limited the precision with which both the return-to-work rate and its predictors could be assessed. This also introduced the risk of non-response bias. Notably, non-respondents were more often rural residents, which may reflect either reluctance or barriers to communication access. In addition, their more favorable clinical profile (absence of angina at low exertion and better exercise capacity) suggests that they may have had better functional status and work capacity. If so, the true rate of return to work may have been underestimated in our results. While this remains speculative, future studies should address missing data more systematically, for example, through sensitivity analyses.
- Enhancethe Discussion by tightening sections less directly related to core findings (e.g., wearable devices).
A: We have tightened the sections of discussion, less directly related to the main findings.
The resting electrocardiogram (ECG) is a simple, widely available tool routinely used in cardiovascular disease (CVD) screening and remains central to the early detection and prevention of adverse cardiac events in the working population [28]. In our study, the absence of ECG abnormalities was a significant predictor of preserved work capacity, highlighting its value not only as a diagnostic marker but also as a functional indicator relevant to occupational outcomes. These findings support the use of ECG in the comprehensive assessment of patients with chronic coronary syndrome, where it may help identify individuals with a higher likelihood of successful reintegration into employment.
- Copyedit to correct typographical errors and improve English usage throughout.
We improved English using the professional editing service provided by MDPI.
Reviewer 3 Report
Comments and Suggestions for Authors
Interesting study, well written. Methodology is good. Results are clearly presented, but they are expected. Discussion is well conducted. Conclusions are in concordance with findings. References are adequate.
Comment: The study included patients with chronic coronary syndrome. How many patients had previous myocardial infarction? How many patients were included in the cardiac rehabilitation program? Was the participation in this program associated with return to work?
Author Response
Dear member of the reviewer board,
Thank you for reviewing our article submitted for publication. We appreciate your insightful comments and have addressed them by making several revisions, as outlined below.
Sincerely,
Dr. Corina Oancea.
Comment: The study included patients with chronic coronary syndrome.
How many patients had previous myocardial infarction?
A: All patients had previous myocardial infarction.
How many patients were included in the cardiac rehabilitation program?
A: None had participated in cardiac rehabilitation prior to their admission to INEMRCM.
Was the participation in this program associated with return to work?
A: Six months after discharge from INEMRCM around 20% of participants had managed to return to work.
Reviewer 4 Report
Comments and Suggestions for Authors
The authors present their manuscript titled "Factors affecting return to work of patients with chronic coronary syndrome. A prospective study." where they report the results of a prospective study of 250 consecutive patients applying for work disability benefits while being diagnosed with chronic coronary syndrome. The patients were followed-up at 6 months regarding whether they have managed to return to work. The main findings consisted of the revealing that sex, resting ECG changes and exercise capacity being predictive factors for work capacity while the type of occupation and the presence of cardiovascular comorbidities are predictive factors for reemployment.
The concept of this study is indeed very interesting since modern therapeutic options in coronary syndromes aim for the eventual reintroduction of the patient to normal life including occupation, especially after participating in cardiovascular rehabilitation. However, there are a few issues that need addressing:
- The article would benefit from improved use of the English language to enhance clarity and readability especially in the Results section. While the scientific content is valuable, certain sections (especially the Results) contain awkward phrasing that may hinder comprehension. I recommend a thorough language edit, possibly with the assistance of a native English speaker or professional editing service.
- The Materials and Methods section would benefit of further being divided into clearly labeled subsections. This would help readers navigate the methodology more easily. Subheading such as "Study Population", "Data collected at baseline", "Exercise testing", "Follow-up" could be useful.
- The Conclusions section does not include any of the correlations and predictive factors that is mentioned in the Results section. I believe it is too general and it would benefit from a more specific discussion of the study’s key findings.
- In the end of the subsection 3.1, the authors report that " In conclusion, we assume that male sex, absence of ECG changes and good exercise level (in METs) may serve as predictors of a high likelihood of employment in case of coronary artery disease". I believe that a better phrasing would be that "In conclusion, we suggest that...". In addition, this part along with the next paragraph is better suited in the Discussion section.
The Results section appears to be significantly impacted by issues with English language usage, which at times obscure the intended meaning and make interpretation of the findings challenging. I recommend careful revision of this section to improve clarity and ensure that the results are communicated accurately and effectively. Engaging a professional language editor or a native English speaker could be especially helpful here.
Author Response
Dear member of the reviewer board,
Thank you for reviewing our article submitted for publication. We appreciate your insightful comments and have addressed them by making several revisions, as outlined below.
Sincerely,
Dr. Corina Oancea.
- The article would benefit from improved use of the English language to enhance clarity and readability especially in the Results section. While the scientific content is valuable, certain sections (especially the Results) contain awkward phrasing that may hinder comprehension. I recommend a thorough language edit, possibly with the assistance of a native English speaker or professional editing service.
A: We improved English using the professional editing service provided by MDPI.
- The Materials and Methods section would benefit of further being divided into clearly labeled subsections. This would help readers navigate the methodology more easily. Subheading such as "Study Population", "Data collected at baseline", "Exercise testing", "Follow-up" could be useful.
A: We introduced subtitles: study design and population with inclusion / exclusion criteria, data collection, exercise testing and follow-up.
- The Conclusions section does not include any of the correlations and predictive factors that is mentioned in the Results section. I believe it is too general and it would benefit from a more specific discussion of the study’s key findings.
We have rewritten the conclusion section including some key findings of our study.
A: In this prospective cohort of patients with chronic coronary syndrome, only one-fifth returned to work within six months, despite nearly half demonstrating preserved functional capacity. Better exercise tolerance, absence of cardiovascular associated pathologies, and engagement in specialized or professional occupations were the main determinants of successful reintegration. Multivariate analysis confirmed occupational specialization and the absence of comorbid cardiovascular diseases as the strongest independent predictors. This findings underscore that return to work after coronary events without adequate support is highly challenging. To improve outcomes, a multidisciplinary approach is necessary, along with the development of effective mechanisms to facilitate employment – such as supportive legislation, clear methodologies, and dedicated rehabilitation structures.
- In the end of the subsection 3.1, the authors report that " In conclusion, we assume that male sex, absence of ECG changes and good exercise level (in METs) may serve as predictors of a high likelihood of employment in case of coronary artery disease". I believe that a better phrasing would be that "In conclusion, we suggest that...". In addition, this part along with the next paragraph is better suited in the Discussion section.
A: We changed “we assume that…” by “The multivariate analysis suggests that…”
We deleted the next paragraph as the comment was already included in the Discussion section.
Round 2
Reviewer 4 Report
Comments and Suggestions for Authors
The authors have adequately answered to my comments. In particular:
- The manuscript is vastly improved language-wise after the authors have used a professional editing service.
- They provided sub-headings for the Materials and Methods section.
- They improved the Conclusions section, making it more specific to the current study.
- They have changed the last section of subsection 3.1 so the term "assume" has been replaced with a more appropriate phrasing.
Author Response
Dear member of the reviewer board,
Thank you for your comments, which helped us improve our article.
Sincerely,
Dr. Corina Oancea.